# Lymph Node and Bone Marrow Micrometastases Define the Prognosis of Patients with pN0 Esophageal Cancer

**DOI:** 10.3390/cancers12030588

**Published:** 2020-03-04

**Authors:** Karl-F. Karstens, Tarik Ghadban, Katharina Effenberger, Guido Sauter, Klaus Pantel, Jakob R. Izbicki, Yogesh Vashist, Alexandra König, Matthias Reeh

**Affiliations:** 1Department of General, Visceral and Thoracic Surgery, University Medical Centre, Hamburg-Eppendorf, Martinistr. 52, 20246 Hamburg, Germany; k.karstens@uke.de (K.-F.K.); t.ghadban@uke.de (T.G.); k.effenberger@uke.de (K.E.); izbicki@uke.de (J.R.I.); yogesh.vashist@gmail.com (Y.V.); alexandra.koenig@klinikum-whv.de (A.K.); 2Department of Pathology, University Medical Centre, Hamburg-Eppendorf, Martinistr. 52, 20246 Hamburg, Germany; g.sauter@uke.de; 3Department of Tumor Biology, University Medical Centre, Hamburg-Eppendorf, Martinistr. 52, 20246 Hamburg, Germany; pantel@uke.de

**Keywords:** esophageal cancer, staging, disseminated tumor cells, lymph node micrometastases

## Abstract

Background: Pathological routine lymph node staging is postulated to be the main oncological prognosticator in esophageal cancer (EC). However, micrometastases in lymph nodes (LNMM) and bone marrow (BNMM) are discussed as the key events in tumor recurrence. We assessed the prognostic significance of the LNMM/BNMM status in initially pN0 staged patients with curative esophagectomy. Methods: From 110 patients bone marrow aspirates and lymph node tissues were analyzed. For LNMM detection immunohistochemistry was performed using the anticytokeratin antibody AE1/AE3. To detect micrometastases in the bone marrow a staining with the pan-keratin antibody A45-B/B3 was done. Results were correlated with clinicopathologic parameters as well as recurrence and death during follow-up time. Results: Thirty-eight (34.5%) patients showed LNMM, whereas in 54 (49.1%) patients BNMM could be detected. LNMM and BNMM positive patients showed a correlation to an increased pT category (*p* = 0.017). Univariate and multivariate analyses revealed that the LNMM/BNMM status and especially LNMM skipping the anatomical lymph node chain were significant independent predictors of overall survival and recurrence-free survival. Conclusions: This study indicates that routine pathological staging of EC is insufficient. Micrometastases in lymph nodes and the bone marrow seem to be the main reason for tumor recurrence and they are a strong prognosticator following curative treatment of pN0 EC.

## 1. Introduction

Esophageal carcinoma (EC) is an aggressive tumor entity with a high rate of local recurrences and distant metastases, even after curative R0 resection in patients without lymph node metastases defined by routine histopathological examination (pN0) [1,2,3]. Although postoperative morbidity and mortality rates have declined over the past decade, survival rates remain poor with a median survival of 16–20 months even after curative treatment [4,5]. Metastatic lymph node invasion is still the strongest prognostic factor for overall survival and tumor-related death following esophagectomy [6]. However, some patients develop tumor recurrence or metastatic disease despite the absence of detectable metastases by routine clinical staging and histopathological examinations at the time of surgery. Thus, an early metastatic relapse of patients after curative treatment in pN0 esophageal cancers suggests that disseminated tumor cells (DTC) or micrometastases might have already been present at the time of preoperative diagnosis [7,8,9,10,11,12,13]. Hence, conventional histopathological examination with eosin and hematoxylin (HE) might not detect single tumor cells or clusters in lymph nodes. These disseminated tumor cells can migrate to lymph nodes or circulate in the peripheral blood to finally reside in secondary organs like the bone marrow. Here, DTCs can stay in dormancy as bone marrow micrometastases (BNMM) and re-circulate in the blood even after years. These cells then become precursors of metastases or local tumor recurrence [14]. Therefore, BNMM and lymph node micrometastases (LNMM) play a pivotal role in cancer progression and tumor recurrence [15]. Detection of disseminated tumor cells in lymph nodes or the bone marrow by an extended initial immunohistopathological analysis might therefore change primarily defined pN0 patients to micrometastases positive patients. Until now, the clinical significance of the metastatic niche represented by disseminated tumor cells in the lymph nodes and bone marrow has not been analyzed in a single cohort of patients, who were treated by surgery only. Thus, we assessed the prognostic significance of the LNMM and BNMM status in regard to the oncological outcome and tumor recurrence of patients following curative esophagectomy without lymph node invasion on routine examination (pN0). 

## 2. Methods

### 2.1. Patients, Study Design, and Inclusion Criteria

After approval by the ethics committee of the chamber of physicians in Hamburg, informed consent was obtained from all patients enrolled in this study. None of the patients had received neoadjuvant or adjuvant (radio-) chemotherapy. To exclude potential biases resulting from different surgical approaches on staging accuracy, only patients with transthoracic en-bloc esophagectomy (TTE) and radical two-field lymphadenectomy were analyzed. Patients who underwent extended surgical procedures, e.g., esophagogastrectomy with subsequent colonic interposition were excluded from this study. Hence, tumor samples, lymph nodes, and bone marrow aspirates of the upper iliac crest were collected in 118 patients. Microscopic tumor staging and grading was performed based on the seventh edition of the tumor-node-metastasis (TNM) classification of the International Union against Cancer. All pathologists were unaware of the immunohistochemical findings in lymph nodes and bone marrow. Only patients without lymph node involvement on routine pathological assessment (pN0) were included in this study. All of these patients had tumor-free resection margins on histopathological examination. Six patients (5.1%) were excluded from analyses, because they died during their hospital stay.

Postoperative follow-up investigations were performed on an outpatient basis in three months intervals after surgery for one year followed by intervals of six months. Follow-up was conducted by physicians, who had no knowledge of the immunohistochemical findings. Two patients, who were alive after 12 and 24 months, were lost to follow-up thereafter and consequently excluded from the study. In total, a cohort of 110 patients was analyzed in this study.

### 2.2. Preparation of the Surgical Specimen and Immunohistochemical Analysis

All lymph nodes were resected during systematic lymphadenectomy and surgically mapped according to the scheme of the American Thoracic Society as modified by Casson et al. [16,17]: (1) regional mediastinal lymph nodes in the vicinity and (2) distant to the tumor; (3) left gastric artery and perigastric lymph nodes, (4) lymph nodes at the celiac trunk; (5) suprapancreatic pancreatic lymph nodes along the common hepatic artery. Lymph nodes were systematically sampled and divided for conventional histopathology, as well as snap-frozen in liquid nitrogen within 3 h after their removal and stored at −80 °C until use. All lymph nodes, which were metastatic free on initial histological workup, were again screened by immunohistochemistry with the anti-human cytokeratin monoclonal antibody AE1/AE3 (IgG1; Dako, Hamburg, Germany) diluted 1:50 by the biotin-streptavidin method as described previously [18]. In some lymph nodes, a faint positivity for cytokeratin was found in small vessels or histiocytes. However, a sample was only considered positive for micrometastasis if a strong cytokeratin-expression was found within single cells or cell-clusters displaying a clear epithelial phenotype (see Figure 1A).

Cryostat sections 5–6 μm thick were cut at three different levels in each node and transferred onto glass slides treated with 3-triethoxysilylpropylamin (Merck, Darmstadt, Germany). One section of the sample obtained at each level was stained by the alkaline phosphatase–antialkaline phosphatase (APAP) technique combined with the new fuchsine stain (Sena, Heidelberg, Germany) to visualize the staining with the AE1/AE3 antibody [19,20].

In 16 control patients with nonepithelial tumors or inflammatory diseases, lymph nodes consistently stained negative. Sections of normal esophageal (AE1/AE3) mucosa served as positive staining controls and isotype-matched, irrelevant murine monoclonal antibodies served as negative controls (purified immunoglobulin mouse myeloma protein for IgG1; Sigma, Deisenhofen, Germany).

Each slide was evaluated in a blinded fashion manner by two pathologists working independently. In the event of inter-observer differences, slides were re-evaluated by a third pathologist and consensual decisions were made. In the event of detection of micrometastases in a lymph node, two adjacent lymph node sections were prepared, stained with hematoxylin and eosin, and screened by pathologists unaware of the preceding findings.

Discontinuous skip metastases were defined as distant nodal micrometastases that did not respect the anatomical lymph node chain [21]. This included a nodal “downstairs” involvement of extra mediastinal, abdominal lymph nodes. In patients with tumor localization below the tracheal bifurcation, an “upstairs” skipping was defined as an involvement of upper mediastinal lymph nodes located above the azygos vein. In addition, at least seven lymph nodes between the outer border of the tumor and the micrometastatic positive lymph needed to be free of micrometastases. To investigate whether patients with micrometastases have different oncological risk profiles, the number of involved nodes was related to the total number of harvested lymph nodes (lymph node ratio). According to Waterman et al. the lymph node ratio was grouped as <10% and ≥10% [22].

Aspirates of 4–8 mL of bone marrow from the iliac crest were taken during primary esophageal cancer surgery. The specimens were collected in heparin, and mononuclear cells, isolated by density-gradient centrifugation through Ficoll–Hypaque (Pharmacia, Freiburg, Germany) at 400× *g* for 30 min, were deposited onto glass slides by cytocentrifugation at 150× *g* for 3 min. To detect tumor cells in bone marrow, we used the monoclonal antibody A45-B/B3 (IgG1; Micromet, Munich, Germany), which has been used to detect BNMM in esophageal cancers previously [10,11,23]. After incubation of the mononuclear cells from each patient with blocking serum (Biotest; diluted 1:10 in PBS) for 20 min, incubation with A45-B/B3 (2 µg/mL in PBS solution with 10% AB serum) was performed. To visualize the staining, the same APAP technique as described for the AE1/AE3 antibody was used (see Figure 1B). Finally, the slides were counterstained with hematoxylin (Mayer’s hemalaun solution, Merck, Germany) and mounted in Kaiser’s glycerine-gelatin (Chroma Gesellschaft Gmbh, Munster, Germany). As an isotype-specific negative control the MOPC-21 monoclonal antibody (Sigma Chemical, St Louis, MO, USA) lacking any known reactivity for epithelial or bone marrow cells, was used at the same concentration as the A45-B/B3 antibody [24].

### 2.3. Statistical Analysis

Patients were grouped according to their LNMM and BNMM status: LNMM and BNMM negative, LNMM positive and BNMM negative, LNMM negative and BNMM positive as well as LNMM and BNMM positive. The LNMM and BNMM negative group was used as control. For group comparison against the control group, the one-way ANOVA test was performed. Associations of categorical variables were evaluated by Fisher’s exact test. The average values are given as median with minimum and maximum as range. For survival analyses, the following primary end points were considered: disease-related death and local or distant recurrence. The survival intervals were computed from the time of surgery to the time of tumor recurrence or to disease-related death. Kaplan–Meier curves were constructed to estimate survival and log-rank test was performed to compare survival variables in univariate analysis. For multivariate analyses a Cox regression was performed calculating the odds ratio (OR) with a 95% confidence interval (95% CI) using the following parameters: age, T category (size and depth of tumor invasion), UICC stage (comprising the TNM categories), grading, presence of skip metastases, lymph node ratio, LNMM and BNMM status. All used tests were two-sided. The cut-off for the significance level was set at 0.05.

### 2.4. Statement of Ethics

The study was approved by the Medical Ethical Committee (approval number: PV3548), Hamburg, Germany. Informed consent was obtained from all patients before study inclusion. All procedures performed in this study involving human participants were in accordance with the ethical standards of the institutional and national research committee and with the 1964 Helsinki declaration and its later amendments or comparable ethical standards.

## 3. Results

### 3.1. Characteristics of Patients and Clinicopathological Parameters

A total of 110 patients (27 female and 83 male) with a median age of 60.9 years (range 36.0–80.0 years) were included. A total of 66 (60.0%) presented with an adenocarcinoma and 44 (40.0%) with a squamous cell carcinoma of the esophagus without lymph node involvement on routine histopathology (pN0), respectively. All patients had consistently undergone transthoracic en-bloc esophagectomy with a median lymph node yield of 23.0 (range 18.0–60.0). A tumor recurrence was found in 39 (35.5%) patients and 25 (22.7%) patients died due to a tumor related cause within the observational period. In detail, the 1-year survival rate was 91.8%, the 3-year survival rate was 80.0%, and 5-year survival rate was calculated to be 77.3%, respectively. Table 1 gives a summary of tumor variables and the characteristics of patients enrolled in the study.

### 3.2. Prevalence of Immunohistochemically Detectable Tumor Cells in Lymph Nodes and Bone Marrow

A total of 2736 lymph nodes, which were tumor free on initial pathological examination, were assayed immunohistochemically. Micrometastases were found in 137 (5.0%) lymph nodes distributed over 38 patients (34.5%). Within these patients a total of 1035 lymph nodes were analyzed. Hence, 13.3% of the lymph nodes of LNMM positive patients presented a micrometastatic infiltration. Micrometastatic tumor load was stratified according to the lymph node ratio ranging from 5.6% to 31.6%. Hence, 25 (22.7%) patients were assigned to a lymph node ratio of ≥10%, while 85 (77.3%) patients were assigned to a lymph node ratio of <10%, accordingly (see Table 2).

Isolated tumor cells in the bone marrow were detected in 54 patients (49.1%). Hence, 41 patients were grouped into LNMM and BNMM negative (control group), 15 patients were assigned to the LNMM positive and BNMM negative group, 31 patients were LNMM negative and BNMM positive, and 23 patients were found to be LNMM and BNMM positive. When comparing the three groups of patients against the control group, LNMM and BNMM positive patients demonstrated a significantly higher rate of disease recurrences as well as a higher rate of tumor related deaths (both *p* < 0.001). In addition, a significantly stronger association with an increased T category was found (*p* = 0.017). None of the other parameters among all groups showed a significant difference compared to LNMM and BNMM negative patients. However, a trend towards more adenocarcinomas was found in the LNMM positive and BNMM negative group (*p* = 0.050). For further details see Table 1.

### 3.3. Association of Micrometastases in Lymph Nodes and Bone Marrow

No direct association was found between the occurrence of lymph node and bone marrow micrometastases (*p* = 0.109). We found a trend between the presence of skip metastases and the occurrence of BNMM indicating more skip metastases in BNMM positive patients. In a further subgroup analysis, the impact of the lymph node ratio on the presence of bone marrow micrometastases was investigated. Interestingly, patients with BNMM were significantly more often found in the group with a higher lymph node ratio as opposed to patients without BNMM (*p* = 0.012). See Table 2.

### 3.4. Survival Analyses

Overall, a disease related death was reported after a median of 20.5 months (range 5.0–88.0 months) and a disease recurrence after a median of 18.5 months (range 3.0–78.0 months). Univariate analysis revealed a significant difference between all four LNMM and BNMM categories in regard to overall and recurrence-free survival (both *p* < 0.001). Patients without any micrometastases (control group) did not reach median overall survival and demonstrated a median recurrence-free survival of 75.0 months. LNMM positive and BNMM negative patients showed a median overall survival of 48.0 months (*p* < 0.001) while LNMM negative and BNMM positive patients did not reach median survival (*p* = 0.034). Of note, LNMM positive and BNMM positive patients demonstrated the shortest overall survival with a median of 17.0 months (*p* < 0.001). In regard to recurrence-free survival LNMM positive and BNMM positive patients had a recurrent disease after a median of 14.0 months (*p* < 0.001). Interestingly, LNMM positive and BNMM negative patients also showed an early recurrence after a median of 13.0 months (*p* < 0.001).

Patients with bone marrow micrometastases only (LNMM negative and BNMM positive) did not reach median recurrence-free survival (*p* = 0.010). To further identify the prognostic impact of LNMM a comparison of the LNMM negative and BNMM positive group with the LNMM and BNMM positive group was performed demonstrating a significant difference in regard to overall und recurrence-free survival (both *p* < 0.001). Accordingly, the prognostic effect of BNMM by comparing the LNMM and BNMM negative group with the LNMM negative and BNMM positive group outlined a significant difference in overall and recurrence-free survival (*p* = 0.034 and *p* = 0.010, respectively). See Figure 2.

Patients with proven skip metastases showed median overall survival rates of 10.0 months and median recurrence-free survival rates of 9.0 months, while patients without skip metastases did not reach median overall survival and had a median recurrence-free survival of 75.0 months (both *p* < 0.001). See Figure 3A. In addition, patients with a lymph node ratio of ≥10% demonstrated a reduced overall (median 16.0 months) and shortened recurrence-free survival (median 11.0 months) as compared to their counterparts (both *p* < 0.001). See Figure 3B.

Multivariate Cox regression analysis identified the presence of lymph node skip metastases (OR 7.953, 95% CI 2.376–26.617; *p* = 0.001) and the micrometastastic spread of tumor cells into the lymph node and bone marrow (OR 1.925, 95% CI 1.126–3.289; *p* = 0.017) as independent risk factors. Interestingly, the lymph node ratio did not reach significance (see Table 3).

In line with the latter findings, recurrence-free survival was closely related to the LNMM and BNMM status as well (OR 1.624, 95% CI 1.085–2.431; *p* = 0.019). Furthermore, skipping of lymph nodes showed a strong association with the occurrence of a tumor recurrence (OR 3.702, 95% CI 1.297–10.571; *p* = 0.014). None of the other parameters demonstrated a significant influence on recurrence-free survival (see Table 4).

## 4. Discussion

Even after curative resection for lymph node negative esophageal cancers, tumor recurrence and development of distant metastases remain an unsolved problem. Undetected disseminated tumor cells in the lymph nodes by routine histopathology workup and or dissemination of tumor cells into the bone marrow might be one possible explanation. Hence, we investigated the spread of these tumor cells towards regional lymph nodes and to the bone marrow simultaneously in patients resected in a curative intent with a lymphadenectomy of at least 18 lymph nodes. All patients demonstrated a pN0 category on conventional histological workup.

We detected 38 (34.5%) patients with LNMM, who were initially staged with pN0 esophageal cancers. These patients presented with a total of 137 (5.0%) lymph node micrometastases. Other studies found 8.0% to 38.0% pN0 esophageal cancer patients with a micrometastatic lymph node involvement and micrometastases in 1.0% to 49.0% of the investigated lymph nodes [8,25,26,27,28,29]. Hence, our data are within the reported range of the literature. Nevertheless, the vast range of results is apparent. One possible explanation might be found in the different sampling and staining techniques for the collected lymph nodes [9,25,30].

We found bone marrow micrometastases in nearly half (49.1%) of the investigated patients. This is in line with other studies reporting BNMM in 21.3% and up to 53.0% of the patients [12,14,31,32,33]. When putting the LNMM and BNMM analyses together, a total of 23 (20.9%) patients with micrometastases in both the lymph nodes and bone marrow were identified. This number is higher as reported by Thorban et al., who performed a subanalysis of 24 squamous cell carcinomas of the esophagus and identified three (12.5%) patients with LNMM and BNMM [32]. However, no further analyses were performed in the latter study due to the limited sample size. In our cohort, no direct correlation between the occurrence of LNMM and BNMM was found. Interestingly, a significantly higher ratio of micrometastatic positive lymph nodes in BNMM positive patients as compared to BNMM negative patients was seen. This might indicate an increased general dissemination of tumor cells and thus more aggressive biological behavior of some tumors. Supporting the latter theory, an increased pT category and advanced UICC stage were more often found in patients, who were LNMM and BNMM positive. Of note, the presence of LNMM (n = 1; 2.6%), BNMM (n = 4; 7.4%), and of both LNMM and BNMM (n = 1; 4.3%) were detected even in pT1a cancers, which due to their confined growth to the esophageal mucosa should be very limited in their ability to disseminate tumor cells. However, this underlines the findings of a recent study reporting that LNMM can already be present in early stage T1a/T1b esophageal carcinomas suggesting an extended lymph node dissection even in these early stages [34].

Interestingly, we detected a trend for a greater metastatic lymph node involvement in the adenocarcinomas subtype. Zingg et al. also did not find a significant difference in the frequency of micrometastases between adenocarcinomas (n = 54) and squamous cell carcinomas (n = 32) of the esophagus [35]. However, they reported that the adenocarcinoma subtype itself was a negative predictor for disease-free survival. Hence, our finding might indicate a more aggressive biological behavior of the latter tumor entity. We showed by univariate and multivariate analyses that the presence of either LNMM or BNMM was an independent prognostic marker for reduced overall and recurrence-free survival. However, the detection of micrometastases in the lymph nodes had the greatest impact on survival since both groups with LNMM presented with the shortest overall and recurrence-free survival. Nevertheless, the prognostic implications of LNMM and BNMM are still being debated. The majority of studies have also found a reduced overall or disease-free survival in LNMM positive patients [8,9,25,28,29]. Heeren et al. investigated 60 initially pN0 staged cancers of the gastroesophageal junction reporting a reduced disease-free survival in patients with a micrometastatic nodal involvement [8]. A separate study found an association of LNMM with the development of local recurrences (*p* = 0.01) and distant metastases (*p* = 0.01). This study also reported a significantly reduced overall survival in LNMM positive patients. However, the latter result was only tested in univariate analysis [9]. In contrast to the latter results, other studies reported no additional prognostic value for LNMM in pN0 staged esophageal cancers [7,22,27]. Hence, the prognostic impact of LNMM remains uncertain. The significance of BNMM on patients’ survival for esophageal cancers is also being debated [12,13,14,31,32]. Ryan et al. analyzed 88 esophageal cancer patients with a follow-up of 10 years showing that BNMM had a significant influence on overall and disease-specific survival especially in patients without neoadjuvant treatment [14]. Two other studies investigating squamous cell carcinomas only also found an independent influence of BNMM on survival [31,32]. On the contrary, Gray et al. reported an association of BNMM with the T category but failed to show a significant impact on survival in univariate and multivariate analyses [33]. In line with the latter results, a recent study investigating 76 early stage esophageal cancers did not find an impact on overall and disease-free survival for BNMM positive patients [13].

Unfortunately, no accepted standard methodology for finding BNMM exists making a direct comparison of studies investigating the presence and effect on patients’ prognosis of BNMM difficult. In this study, we used cytokeratin immunohistochemistry, which is used as a standard method in most pathology laboratories for the detection of metastatic carcinomas [36,37]. Additionally, it has been shown for esophageal cancers and other cancer entities that cytokeratin-positive cells in the bone marrow are tumor cells [38,39,40]. Another advantage of immunocytochemical assays is the absence of cut-off limitations of the targeted marker as opposed to polymerase chain reaction-based methods [41,42]. We obtained the bone marrow aspirates from the iliac crest. However, other authors prefer to take samples from the rib cage since some studies have shown a superiority in detecting micrometastases for esophageal cancers at this sample site [43,44,45]. In addition, other potential confounders like the sample size, different neo-adjuvant treatments, and also different surgical techniques may be one explanation for the range of results regarding the prognostic effect of LNMM and BNMM in upper gastrointestinal cancers. In this study, we focused on radiochemotherapy naïve patients to rule out a potential bias on disseminated tumor cells. However, a separate study indicates that neo-adjuvant radiochemotherapy reduces the rate of LNMM in patients with esophageal squamous cell carcinomas and improves survival [46].

Nevertheless, our results emphasize the prognostic role of lymph node and bone marrow micrometastases in early stage esophageal cancers. Especially since we did not find a correlation between pT category or UICC stage with patients’ survival on multivariate analyses. This stresses the need for additional staging options in this subset of esophageal cancers and thereby implicates a routine additional immunohistochemical staining, which should be performed regardless of the pT category.

We also looked into the pattern of the micrometastatic lymph node spread. Therefore, discontinuous skip metastases were defined as distant nodal micrometastases that did not respect the anatomical lymph node chain. Interestingly, we were able to show that a discontinuous lymph node spread of tumor cells was more likely associated with BNMM as compared to a continuous lymph node involvement respecting the anatomic levels. In addition, the presence of skip metastases was significantly associated with reduced overall and recurrence-free survival. Skip metastases even were the strongest investigated parameter for overall and recurrence-free survival indicating an already advanced disease. Interestingly, despite being a significant prognosticator in univariate analysis the relative amount of lymph nodes positive for micrometastases failed to show an independent association with overall and recurrence-free survival on multivariate analysis. Hence, the latter finding underlines that not the amount of metastatic lymph nodes but rather the biological behavior of the disseminated tumor cells determines patients’ prognosis. Therefore, an extended lymphadenectomy should be performed on a routine basis to improve tumor staging and evaluate a possible adjuvant chemotherapy even in early stage esophageal cancers. 

## 5. Conclusions

In conclusion, the occult systemic spread of micrometastases to the lymph nodes or bone marrow at time of primary diagnosis has a significant effect on survival in pN0 esophageal cancers. Especially, the occurrence of lymph node micrometastases with a skipping of the anatomical node chain is associated with earlier disease related death and recurrence. In addition, skip metastases tend to occur more often in BNMM positive patients. Hence, initially pN0 staged patients would benefit from an extended lymphadenectomy and an immunohistochemical workup to improve staging accuracy and thereby facilitate a proper adjuvant treatment.

## Figures and Tables

**Figure 1 cancers-12-00588-f001:**
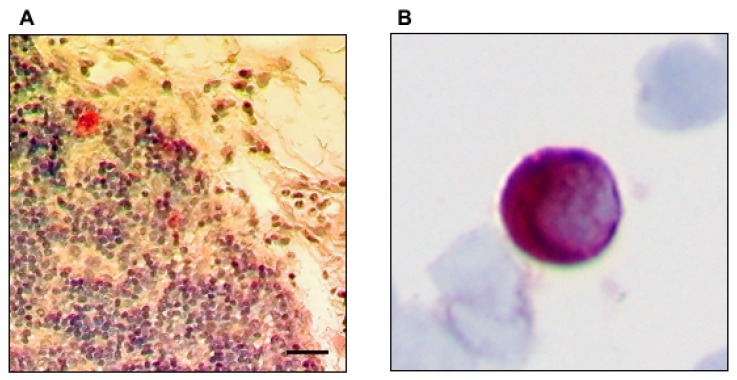
Representative images of the immunohistochemical staining for lymph node micrometastases (LNMM) and bone marrow micrometastases (BNMM). (**A**) Cytokeratin positive cells labeled by AE1/AE3 in the lymph node. Bar indicates 20 µm. (**B**) Cytokeratin positive cells in the bone marrow labeled by A45-B/B3 in 600-fold magnification.

**Figure 2 cancers-12-00588-f002:**
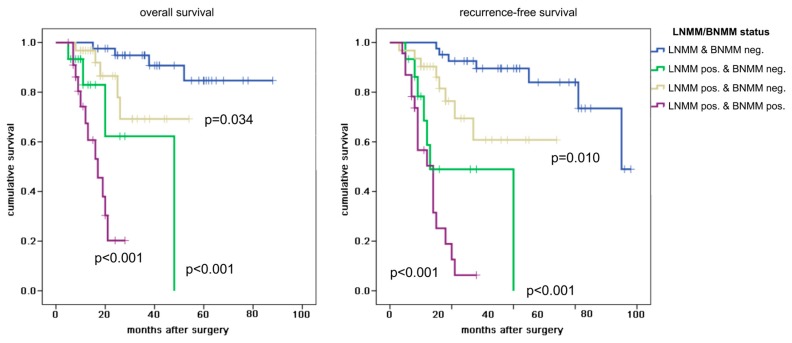
Kaplan–Meier curves of the lymph node and bone marrow micrometastases status for overall and recurrence-free survival. LNMM/BNMM status: lymph node and bone marrow metastases status (LNMM and BNMM negative, LNMM positive and BNMM negative, LNMM negative and BNMM positive, LNMM and BNMM positive); log-rank test was performed against LNMM and BNMM negative. The crosses mark censored patients. *p* value < 0.05 is considered significant.

**Figure 3 cancers-12-00588-f003:**
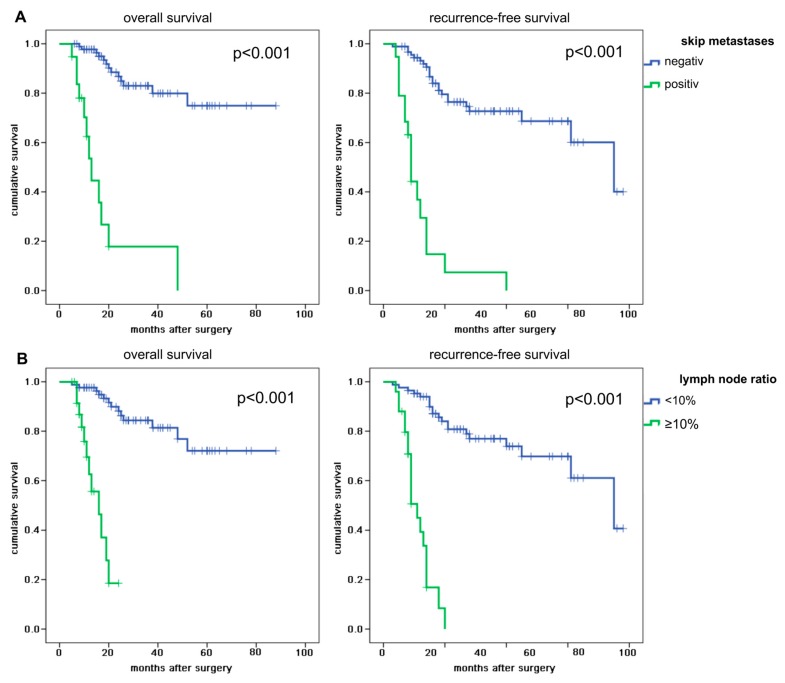
Kaplan–Meier curves of the presence of skip metastases and lymph node ratio for overall and recurrence-free survival. (**A**) Kaplan–Meier curves of the presence of skip metastases for overall and recurrence-free survival. (**B**) Kaplan–Meier curves of the lymph node ratio for overall and recurrence-free survival. The crosses mark censored patients. *p* value < 0.05 is considered significant.

**Table 1 cancers-12-00588-t001:** Comparison of the lymph node/bone marrow micrometastases status with patients’ clinicopathological characteristics.

Variables	All	LNMM and BNMM Negative	LNMM Positive and BNMM Negative	*p* Value	LNMM Negative and BNMM Positive	*p* Value	LNMM and BNMM Positive	*p* Value
Patients	110 (100.0%)	41 (37.3%)	15 (13.6%)	n.a.	31 (28.2%)	n.a.	23 (20.9%)	n.a.
Age (years)								
≤60	53 (48.2%)	21 (51.2%)	8 (53.3%)	0.999	13 (41.9%)	0.868	11 (47.8%)	0.994
>60	57 (51.8%)	20 (48.8%)	7 (46.7%)		18 (58.1%)		12 (52.2%)	
Sex								
Male	83 (75.5%)	33 (80.5%)	9 (60.0%)	0.398	22 (71.0%)	0.790	19 (82.6%)	0.998
Female	27 (24.5%)	8 (19.5%)	6 (40.0%)		9 (29.0%)		4 (17.4%)	
pT category								
pT1	24 (21.8%)	13 (31.7%)	2 (13.3%)	0.112	8 (25.8%)	0.938	1 (4.3%)	*0.017*
pT2	35 (31.8%)	13 (31.7%)	3 (20.0%)		12 (38.7%)		7 (30.4%)	
pT3	37 (33.6%)	13 (31.7%)	7 (46.7%)		7 (22.6%)		10 (43.5%)	
pT4	14 (12.7%)	2 (4.9%)	3 (20.0%)		4 (12.9%)		5 (21.7%)	
UICC stage								
I	24 (21.8%)	13 (31.7%)	2 (13.3%)	0.175	8 (25.8%)	0.975	1 (4.3%)	*0.011*
II	72 (65.5%)	28 (63.4%)	10 (66.7%)		19 (61.3%)		17 (73.9%)	
III	14 (12.7%)	2 (4.9%)	3 (20.0%)		4 (12.9%)		5 (21.7%)	
Grading								
G1	11 (10.0%)	6 (14.6%)	1 (6.7%)	0.516	2 (6.5%)	0.988	2 (8.7%)	0.970
G2	65 (59.1%)	23 (56.1%)	7 (46.7%)		21 (67.7%)		14 (60.9%)	
G3	34 (30.9%)	12 (29.3%)	7 (46.7%)		8 (25.8%)		7 (30.4%)	
Histology								0.351
AC	66 (60.0%)	20 (48.8%)	13 (86.7%)	0.050	17 (54.8%)	0.952	16 (69.6%)	
SCC	44 (40.0%)	21 (51.2%)	2 (13.3%)		14 (45.2%)		7 (30.4%)	
Recurrence								
Negative	71 (64.5%)	34 (82.9%)	8 (53.3%)	0.113	23 (74.2%)	0.832	6 (26.1%)	*<0.001*
Positive	39 (35.5%)	7 (17.1%)	7 (46.7%)		8 (25.8%)		17 (73.9%)	
Survival								
Alive	85 (77.3%)	37 (90.2%)	11 (73.3%)	0.489	26 (83.9%)	0.905	11 (47.8%)	*<0.001*
Deceased	25 (22.7%)	4 (9.8%)	4 (26.7%)		5 (16.1%)		12 (52.2%)	

LNMM: lymph node micrometastases; BNMM: bone marrow micrometastases; UICC: Union for International Cancer Control; AC: adenocarcinoma; SCC: squamous cell carcinoma; “LNMM and BNMM negative” serves as control group and the other groups are tested against it. *p* value < 0.05 is considered significant; n.a.: not applicable; significant values are highlighted in italic.

**Table 2 cancers-12-00588-t002:** Associations between lymph node and bone marrow micrometastases.

Variables	BNMM		*p* Value
	Negative	Positive	
LNMM			
Negative	41 (73.2%)	31 (57.4%)	0.109
Positive	15 (26.8%)	23 (42.6%)	
Skip metastases			
Negative	50 (89.3%)	41 (75.9%)	0.080
Positive	6 (10.7%)	13 (24.1%)	
Lymph node ratio			
<10%	49 (87.5%)	36 (66.7%)	*0.012*
≥10%	7 (12.5%)	18 (33.3%)	

LNMM: lymph node micrometastases; BNMM: bone marrow micrometastases; *p* value < 0.05 is considered significant; significant values are highlighted in italic.

**Table 3 cancers-12-00588-t003:** Multivariate analysis for overall survival.

Variables	OR	95% CI	*p* Value
Age	1.011	0.963–1.062	0.778
pT category	1.284	0.440–3.742	*0.647*
UICC stage	2.144	0.366–12.201	0.403
Grading	1.347	0.610–2.975	0.461
Skip metastases	7.953	2.376–26.617	*0.001*
Lymph node ratio	1.204	0.246–5.887	0.819
LNMM/BNMM status	1.925	1.126–3.289	*0.017*

OR (odds ratio), CI (confidence interval). Age in years; pT: pathological T category; UICC: cancer staging according to the Union for International Cancer Control; Grading: pathological grading; skip metastases: absence or presence of micrometastases ignoring the anatomical lymph node chain; lymph node ratio: amount of micrometastases in relation to the total number of harvested lymph nodes (<10% or ≥10%); LNMM/BNMM status: lymph node and bone marrow metastases status (LNMM and BNMM negative, LNMM positive and BNMM negative, LNMM negative and BNMM positive, LNMM and BNMM positive); *p* value < 0.05 is considered significant; significant values are highlighted in italic.

**Table 4 cancers-12-00588-t004:** Multivariate analysis for recurrence-free survival.

Variables	OR	95% CI	*p* Value
**Age**	0.991	0.952–1.031	0.642
**pT category**	1.319	0.560–3.103	*0.527*
**UICC stage**	2.119	0.524–8.580	0.292
**Grading**	1.092	0.597–1.997	0.775
**Skip metastases**	3.702	1.297–10.571	*0.014*
**Lymph node ratio**	2.686	0.758–9.524	0.126
**LNMM/BNMM status**	1.624	1.085–2.431	*0.019*

OR (odds ratio), CI (confidence interval). Age in years; pT: pathological T category; UICC: cancer staging according to the Union for International Cancer Control; Grading: pathological grading; skip metastases: absence or presence of micrometastases ignoring the anatomical lymph node chain; lymph node ratio: amount of micrometastases in relation to the total number of harvested lymph nodes (<10% or ≥10%); LNMM/BNMM status: lymph node and bone marrow metastases status (LNMM and BNMM negative, LNMM positive and BNMM negative, LNMM negative and BNMM positive, LNMM and BNMM positive); *p* value < 0.05 is considered significant; significant values are highlighted in italic.

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
