# Peer review of "Lymph Node and Bone Marrow Micrometastases Define the Prognosis of Patients with pN0 Esophageal Cancer"

_cancers, 2020, doi:10.3390/cancers12030588_

Round 1

Reviewer 1 Report

This is an exceptionally well conducted and planned study about lymph node and bone marrow mircometastases in resectable esophageal cancer. The authors have made decent effort in order to exclude potential sources of bias by excluding N+ tumors and patients with unsatistactory lymphadenectomy. Therefore the results can be considered reliable.

I have only minor comments:

Introduction line 52: “These patients would benefit from an adjuvant treatment, respectively.” Add reference or remove the sentence.

Results: Age with single digit

Table 2: Only stage I and II are listed although you have 14 T4 tumors?

You have exceptionally good survival (77.3%). In the results section you could add Kaplan-meier based estimates of 1-, 3- and 5-year survival of all 110 patients in order to put this in context.

You must improve all Figures. You can draw these with SPSS, but at least remove the gray background and improve the resolution.

Discussion: You have some repetition of the results in Discussion section. For example last paragraph before conclusion, you repeat ORs which is not necessary.

In discussion section last paragraph, add proper clinical suggestions based on your study. Do you consider that micrometastases or bone marrow metastases should be searched from all patients? Recardless of tumor T stage?

Author Response

This is an exceptionally well conducted and planned study about lymph node and bone marrow mircometastases in resectable esophageal cancer. The authors have made decent effort in order to exclude potential sources of bias by excluding N+ tumors and patients with unsatistactory lymphadenectomy. Therefore the results can be considered reliable.

I have only minor comments:

Introduction line 52: “These patients would benefit from an adjuvant treatment, respectively.” Add reference or remove the sentence.

We removed the sentence from the introduction since it rather belongs into the discussion section. Thus, we state the following in the conclusion “Hence, initially pN0 staged patients would benefit from an extended lymphadenectomy and an immunohistochemical workup to improve staging accuracy and thereby facilitate a proper adjuvant treatment.“ (line 617 – 619, page 10)

Results: Age with single digit

This has been adjusted accordingly and the age is now stated as “…of 60.9 years.” (line 147, page 4).

Table 2: Only stage I and II are listed although you have 14 T4 tumors?

We thank the reviewer for this remark. Indeed, we have 14 T4 tumors, which certainly belong into a UICC III group. We have adjusted Table 1 and the multivariate analyses accordingly. Now the UICC stage was significantly different in patients with LNMM and BNMM positivy as compared to controls (p=0.011). However, in multivariate analyses T stage didn’t show a significant influence on survival anymore. Thus, the sentences describing the T stage findings for survival were removed from the result section. None of the other significances changed.

You have exceptionally good survival (77.3%). In the results section you could add Kaplan-meier based estimates of 1-, 3- and 5-year survival of all 110 patients in order to put this in context.

To give the requested additional information about survival rates, we added the following sentence to the result section: “In detail, the 1-year survival rate was 91.8%, the 3-year survival rate was 80.0% and the 5-year survival rate was calculated to be 77.3%, respectively.” (line 168 – 169; page 4)

You must improve all Figures. You can draw these with SPSS, but at least remove the gray background and improve the resolution.

The reviewer is right about the lack of clarity in the figures. We have adjusted the figures to achieve a better visualization and added the following sentence to the legend: “The crosses mark censored patients.“.

Discussion: You have some repetition of the results in Discussion section. For example last paragraph before conclusion, you repeat ORs which is not necessary.

We removed the OR from the discussion section (line 604 and 605; page 10) to avoid repetition.

In discussion section last paragraph, add proper clinical suggestions based on your study. Do you consider that micrometastases or bone marrow metastases should be searched from all patients? Recardless of tumor T stage?

We thank the reviewer for this comment, which will improve the reading of the manuscript. We added the following paragraph to the discussion, based on the new results of the multivariate analyses: Nevertheless, our results emphasize the prognostic role of lymph node and bone marrow micrometastases in early stage oesophageal cancers. Especially since we did not find a correlation between pT category or UICC stage with patients’ survival on multivariate analyses. This stresses the need for additional staging options in this subset of oesophageal cancers and thereby implicates a routine additional immunohistochemical staining, which should be performed regardless of the pT category.” (line 596 – 600; page 10)

Reviewer 2 Report

Review of manuscript for the Journal Cancer

Title of manuscript: Lymph node and bone marrow micrometastases define the prognosis of patients with pN0 oesophageal cancer.

Karl-Frederick Karstens, Tarik Ghadban, Katharina Effenberger, Guido Sauter, Klaus Pantel, Jakob Izbicki, Yogesh Vashist, Alexandra König,  Matthias Reeh.

This is an interesting study – which is addressing the high rate of recurrence in oesophageal cancer patients who would normally be classified as lymph node negative (pN0). The detection of micro-metastatic cells in the lymph nodes and bone marrow by immunohistochemistry was an independent predictor of survival.

Comments to be addressed by the authors.

It is not clear to me why two different antibodies were used for the two different locations.

Do you need two locations or just two different antibodies?

I assume this was a technical reason – related to background environment – but this is not stated. – i.e. – if they had just stained the all lymph nodes with the AE1/AE3 anti-body and the A45-B/B3 pan cytokeratin antibody – would all the same patients have been picked up?

Were the primary tumors stained with these anti-bodies – to ensure expression?

There are no images of the staining which is disappointing. It is stated that a sample is considered positive if strong cytokeratin-expression was found within cells or cell-clusters displaying a clear epithelial phenotype. It would be good for readers to know what a positive score looks like in both tissue types.

In figures 1 and 2 – it is impossible to make out the colors and text on the right hand side. This text will have to be clarified – or deleted and the meaning of the line colors added to the legend below.

Author Response

This is an interesting study – which is addressing the high rate of recurrence in oesophageal cancer patients who would normally be classified as lymph node negative (pN0). The detection of micro-metastatic cells in the lymph nodes and bone marrow by immunohistochemistry was an independent predictor of survival.

 Comments to be addressed by the authors.

It is not clear to me why two different antibodies were used for the two different locations.

Do you need two locations or just two different antibodies?

I assume this was a technical reason – related to background environment – but this is not stated. – i.e. – if they had just stained the all lymph nodes with the AE1/AE3 anti-body and the A45-B/B3 pan cytokeratin antibody – would all the same patients have been picked up?

We thank the reviewer for this remark. Indeed, a higher background noise was the initial reason to establish staining protocols with two different antibodies. These stainings have also been used in previous publications of our clinic with good results. Hence, we continued using these methods.

 Were the primary tumors stained with these anti-bodies – to ensure expression?

The primary tumors were not stained with these antibodies. However, we used negative controls to ensure a true negative staining.

 There are no images of the staining which is disappointing. It is stated that a sample is considered positive if strong cytokeratin-expression was found within cells or cell-clusters displaying a clear epithelial phenotype. It would be good for readers to know what a positive score looks like in both tissue types.

The reviewer is right about the lack of images of the cytokeratin staining. To fill this gap we have added an additional figure showing the staining for lymph node and bone marrow micrometastases (see below).

Figure 1

Representative images of the immunohistochemical staining for LNMM and BNMM

  1. A) Cytokeratin positive cells labeled by AE1/AE3 in the lymph node. Bar indicates 20µm. B) Cytokeratin positive cells in the bone marrow labeled by A45-B/B3 in 600-fold magnification.

In figures 1 and 2 – it is impossible to make out the colors and text on the right hand side. This text will have to be clarified – or deleted and the meaning of the line colors added to the legend below.

The reviewer is right about the lack of clarity in the figures. We have adjusted the figures to achieve a better visualization and added the following sentence to the legend: “The crosses mark censored patients.“.

Reviewer 3 Report

"Lymph node and bone marrow micrometastases define the prognosis of patients with pN0 oesophageal cancer" by Karstens et al provides a systematic study of the clinical significance of bone marrow micrometastases in oesophageal cancer. The manuscript is very well written and provides excellent background and context for understanding the study. The discussion is particularly thought-provoking and provides an excellent overview of the literature in this field. I only have a few minor (optional) requests. It would be very nice to provide an example image of micrometastases, particularly as Cancers is read by both basic and clinical cancer scientists and this could provide a reference for basic researchers examining micrometastases in animal models. Was the selection of A45-B/B3 for detection of BNMM instead of AE1/AE3 based purely on prior use in the literature or is there a compelling technical rationale for this selection?

Author Response

"Lymph node and bone marrow micrometastases define the prognosis of patients with pN0 oesophageal cancer" by Karstens et al provides a systematic study of the clinical significance of bone marrow micrometastases in oesophageal cancer. The manuscript is very well written and provides excellent background and context for understanding the study. The discussion is particularly thought-provoking and provides an excellent overview of the literature in this field. I only have a few minor (optional) requests. It would be very nice to provide an example image of micrometastases, particularly as Cancers is read by both basic and clinical cancer scientists and this could provide a reference for basic researchers examining micrometastases in animal models. Was the selection of A45-B/B3 for detection of BNMM instead of AE1/AE3 based purely on prior use in the literature or is there a compelling technical rationale for this selection?

The reviewer is right about the lack of images of the cytokeratin staining. To fill this gap we have added an additional figure showing the staining for lymph node and bone marrow micrometastases (see above). Indeed, a higher background noise was the initial reason to establish staining protocols with two different antibodies. These stainings have also been used in previous publications of our clinic with good results [1,2]. Hence, we continued using these methods.

[1]   Koenig AM, Prenzel KL, Bogoevski D, Yekebas EF, Bubenheim M, Faithova L, et al. Strong impact of micrometastatic tumor cell load in patients with esophageal carcinoma. Ann Surg Oncol 2009;16:454–62. https://doi.org/10.1245/s10434-008-0169-7.

[2]   Vashist YK, Effenberger KE, Vettorazzi E, Riethdorf S, Yekebas EF, Izbicki JR, et al. Disseminated tumor cells in bone marrow and the natural course of resected esophageal cancer. Ann Surg 2012;255:1105–12. https://doi.org/10.1097/SLA.0b013e3182565b0b.

Reviewer 4 Report

The authors of "Lymph node and bone marrow micrometastases define the prognosis of patients with pN0 oesophageal cancer" investigated the effect of lymph node and bone marrow micrometastases on overall survival and recurrence-free survival of patients with oesophageal cancer who underwent curative esophagectomy and whose lymph nodes were found to be free of metastases. They found micrometastases in lymph nodes and the bone marrow to be the linked with tumour recurrence. The importance of discontinuous skip metastases was investigated as well and their presence was associated with earlier disease related death and recurrence. These findings are compelling and relevant.

I do have some comments/suggestions to improve the manuscript further:

Abstract: There is no mention of the skip metastases in the abstract, although a lot of value is assigned to them in the discussion and conclusion. Therefore, I would make sure to mention these in the abstract.

Introduction: In general, the introduction should more clearly state the aim(s) of the study and the novelty of the performed research. In more detail, line 36 mentions 5-year survival rates, please define and reference. Line 52 mentions adjuvant treatment, please elaborate and reference.

Methods: Line 88 - Staining protocol details for cytokeratin antibody are missing. Line 94 - The explanation of what the alkaline phosphatase– antialkaline phosphatase (APAP) technique is used for is missing. Line 107 - A reference from the literature for discontinuous skip metastases is missing. Line 120 - Staining protocol details for A45-B/B3 antibody are missing, as well as positive/negative controls. Line 134 - A definition for T category and UICC stage is missing.

Results: In general, figures 1 and 2 are of too poor resolution to be able to be interpreted and need to be replaced with higher resolution charts. Smaller remarks: I believe line 207 holds a mistake in p value (p<0.034 should probably be p=0.034) and line 232 holds a discrepancy as well (OR 1.662 does not match the value in Table 4).

Discussion: The paragraph starting at line 333 discusses the importance of skip metastases. Is there no literature available to compare your results to?

Language: Line 53 - "respectively" is misplaced here, please correct. Line 113 - "a profiles", please choose singular or plural. Line 233 - please correct "association on" to "association with". Line 286 - please correct "this" to "these early stages". Line 296 - please find better wording for "though", e.g. nevertheless. Line 296 - please correct "is" to "are" (limitations ARE being debated).  Line 303 - please correct "on the contrast to" to "in contrast to". Line 310 -"an associations", please choose singular or plural. Line 349 - "spread by micrometastases"; do you mean to say "spread by micrometastasis or "spread of micrometastases"? Line 351 - please remove comma after "especially". 

Author Response

Abstract: There is no mention of the skip metastases in the abstract, although a lot of value is assigned to them in the discussion and conclusion. Therefore, I would make sure to mention these in the abstract.

We thank the reviewer for this comment and edited the introduction as follows:Univariate and multivariate analyses revealed that the LNMM/BNMM status and especially LNMM skipping the anatomical lymph node chain were significant independent predictors of overall survival and recurrence-free survival. “ (line 24 – 27; page 1)

Introduction: In general, the introduction should more clearly state the aim(s) of the study and the novelty of the performed research. In more detail, line 36 mentions 5-year survival rates, please define and reference. Line 52 mentions adjuvant treatment, please elaborate and reference.

We changes the sentence as follows and added the requested references: “Although postoperative morbidity and mortality rates have declined over the past decade, survival rates remain poor with a median survival of 16 to 20 months even after curative treatment [4,5]“. (line 36 -37; page 1)

We removed the sentence regarding the adjuvant treatment from the introduction since it rather belongs into the discussion section (line 52).

Methods: Line 88 - Staining protocol details for cytokeratin antibody are missing. Line 94 - The explanation of what the alkaline phosphatase– antialkaline phosphatase (APAP) technique is used for is missing. Line 107 - A reference from the literature for discontinuous skip metastases is missing. Line 120 - Staining protocol details for A45-B/B3 antibody are missing, as well as positive/negative controls. Line 134 - A definition for T category and UICC stage is missing.

We thank the reviewer for the remarks and added the following sentences to the method section to further explain the staining methods: ”All lymph nodes, which have been metastatic free on initial histological workup, were again screened by immunohistochemistry with the anti-human cytokeratin monoclonal antibody AE1/AE3 (IgG1; Dako, Hamburg, Germany) diluted 1:50 by the biotin-streptavidin method as described previously [5].“ (line 88 -91; page 2) and „ After incubation of the mononuclear cells from each patient with blocking serum (Biotest; diluted 1:10 in PBS) for 20 minutes, incubation with A45-B/B3 (2µg/ml in PBS solution with 10% AB serum) was performed. To visualize the staining, the same APAP technique as described for the AE1/AE3 antibody was used. Finally, the slides were counterstained with hematoxylin (Mayer's hemalaun solution, Merck, Germany) and mounted in Kaiser's glycerinegelatin (Chroma Gesellschaft Gmbh, Munster, Germany). As an isotype-specific negative control the MOPC-21 monoclonal antibody (Sigma Chemical, St Louis, MO, USA) lacking any known reactivity for epithelial or bone marrow cells, was used at the same concentration as the A45-B/B3 antibody (24). (line 127 – 132; page 3).

Regarding the usage of the APAP technique we added the following sentence: “One section of the sample obtained at each level was stained by the alkaline phosphatase–antialkaline phosphatase (APAP) technique combined with the new fuchsine stain (Sena, Heidelberg, Germany) to visualize the staining with the AE1/AE3 antibody [7,8].” (line 99 – 101; page 3)

A reference regarding the definition of skip metastases (line 113; page 3) and a definition of the T category and UICC stage were added. “…age, T category (size and depth of tumour invasion), UICC stage (comprising the TNM categories), grading...“(line 142 – 143; page 3)

Results: In general, figures 1 and 2 are of too poor resolution to be able to be interpreted and need to be replaced with higher resolution charts. Smaller remarks: I believe line 207 holds a mistake in p value (p<0.034 should probably be p=0.034) and line 232 holds a discrepancy as well (OR 1.662 does not match the value in Table 4).

The reviewer is right about the lack of clarity in the figures. We have adjusted the figures to achieve a better visualization and added the following sentence to the legend: “The crosses mark censored patients.“.

In addition, the errors in the results section regarding the p value and OR were fixed (line 224; page 6 and line 248; page 7).

Discussion: The paragraph starting at line 333 discusses the importance of skip metastases. Is there no literature available to compare your results to?

We performed an additional search of the literature but did not find any studies describing micrometastatic skip metastases in oesophageal cancers. However, conventional skip metastases are described in the literature. Nevertheless, we find it not reasonable to compare the rate of conventional lymph node metastases with the rate of lymph node micrometastases.

Language: Line 53 - "respectively" is misplaced here, please correct. Line 113 - "a profiles", please choose singular or plural. Line 233 - please correct "association on" to "association with". Line 286 - please correct "this" to "these early stages". Line 296 - please find better wording for "though", e.g. nevertheless. Line 296 - please correct "is" to "are" (limitations ARE being debated).  Line 303 - please correct "on the contrast to" to "in contrast to". Line 310 -"an associations", please choose singular or plural. Line 349 - "spread by micrometastases"; do you mean to say "spread by micrometastasis or "spread of micrometastases"? Line 351 - please remove comma after "especially".

We thank the reviewer for thoroughly reading the manuscript and corrected the above-mentioned errors accordingly.